# Style Adaptation and Uncertainty Estimation for Multi-Source Blended-Target Domain Adaptation

**Yuwu Lu,**[*] **Haoyu Huang, and Xue Hu**
School of Artificial Intelligence, South China Normal University
{luyuwu2008, hyhuang99, hx1430940232}@163.com

## Abstract

Blended-target domain adaptation (BTDA), which implicitly mixes multiple sub-target domains into a fine domain, has attracted more attention in recent years. Most previously developed BTDA approaches focus on utilizing a single source domain, which makes it difficult to obtain sufficient feature information for learning domain-invariant representations. Furthermore, different feature distributions derived from different domains may increase the uncertainty of models. To overcome these issues, we propose a style adaptation and uncertainty estimation (SAUE) approach for multi-source blended-target domain adaptation (MBDA). Specifically, we exploit the extra knowledge acquired from the blended-target domain, where a similarity factor is adopted to select more useful target style information for augmenting the source features. Then, to mitigate the negative impact of the domain-specific attributes, we devise a function to estimate and mitigate uncertainty in category prediction. Finally, we construct a simple and lightweight adversarial learning strategy for MBDA, effectively aligning multi-source and blended-target domains without the requirements of domain labels of the target domains. Extensive experiments conducted on several challenging DA benchmarks, including the ImageCLEF-DA, Office-Home, VisDA 2017, and DomainNet datasets, demonstrate the superiority of our method over the state-of-the-art (SOTA) approaches.

## 1 Introduction

Domain adaptation (DA), whose objective is to transfer knowledge from one or more well-labeled source domains to a non-labeled target domain, has been studied in recent years [1, 2, 3, 4, 5, 6], including object classification [1, 2], semantic segmentation [3, 4], and object detection [5]. However, most DA approaches are based on a setting that has single source domain and single target domain [1, 2]. In the real world, unlabeled target domains are usually drawn from different distributions. Therefore, most of the existing single target-based DA approaches may not be the best answer to address domain shifts in reality.

Fortunately, an increasing number of researchers have focused on the above-mentioned issues, and multi-target domain adaptation (MTDA) [7, 8, 9] has been studied. MTDA aims to learn a model that can simultaneously utilize information from single source domain and multiple target domains and then perform well in multiple target domains. For instance, HGAN [10] adopts a heterogeneous graph attention network to explore the relations among multiple target domain features. In [11], multiple expert models employed the corresponding source-target domain pair-groups and were then aligned by a student model. Although the existing MTDA approaches have made certain progress, the standard MTDA is still facing challenges because massive amounts of unlabeled data drawn from different distributions are commonly used in real-world settings. It is time-consuming and expensive to divide massive data into multiple corresponding distributions (target domains). For example, for

---

[*]corresponding author

38th Conference on Neural Information Processing Systems (NeurIPS 2024).

Table 1: Comparison about different settings of DA.

| Settings | Source domain number | Target domain number | Domain labels |
|----------|----------------------|----------------------|---------------|
| UDA/SSDA | single | single | ✓ |
| MSDA | multiple | single | ✓ |
| MTDA | single | multiple | ✓ |
| BTDA | single | multiple | × |
| MMDA | multiple | multiple | ✓ |
| MBDA | multiple | multiple | × |

encrypted data stored in a cloud server, due to privacy protection, models cannot directly know the origins of these data (domain labels), which are drawn from different distributions and are blended into a large target domain. Based on the above-mentioned case, blended-target DA (BTDA), which is a more beneficial scenario in real-world settings, has been proposed [12].

Current BTDA approaches [12, 13] are mainly based on three points: 1) the adaptation process contains single source domain and multiple target domains. 2) During the adaptation process, the model cannot access both the domain labels and category labels of the target domains. 3) In blended-target domain, the category labels of each sub-target domain may not follow the same distribution. Therefore, simply utilizing MTDA or SSDA (single source domain adaptation) methods to handle the BTDA task may lead to negative transfer because the domain labels of target domains are unseen, and the category feature space is a hybrid space. As the first deep learning work focused on the BTDA scenario, AMEAN [12] employs two adversarial learning steps and utilizes meta-learning to minimize the domain gap between the source domain and the blended-target domain. However, insufficient information obtained from single source domain makes models difficult to align the distributions of multiple target domains. Moreover, the presence of different distributions in the blended-target domain may aggravate negative transfer. Recently, multi-source domain adaptation (MSDA) [14, 15, 16] has produced impressive results. MSDA approaches can utilize more feature information from extra source domains to learn domain-invariant representations, effectively solving negative transfer. However, as far as we know, no related works have been proposed to utilize more feature information from multiple source domains for BTDA.

In this paper, to further exploit feature information from multiple domains, we pay attention to the BTDA in the case of multiple source domains, *i.e.,* Multi-Source Blended-Target Domain Adaptation (MBDA). The comparisons of different DA settings are illustrated in Table 1. At the same time, a style adaptation and uncertainty estimation (SAUE) method is proposed for MBDA. Different from previous works, we utilize the style information of the blended-target domain to enhance source domain features, thus building a better representation space. Specifically, we first simultaneously extract the source and target style information and then calculate the similarity factors between the source and target style information. The similarity factors are served as the weighted matrix during the feature augmentation process. Based on style adaptation, we further analyze the uncertainty in the classification model and adopt a loss function to eliminate the uncertainty introduced by the multi-source domains. In addition, as the domain labels of the blended-target domain are unavailable in MBDA, we construct an adversarial learning scheme for MBDA without the requirement of domain labels of the target domains.

The main contributions of this work are summarized as follows:

- An approach SAUE is proposed to explore information from multiple source domains for BTDA. As far as we know, SAUE is the first work that was proposed for MBDA, which can utilize more feature information from extra source domains to learn domain-invariant representations.

- To further exploit the style information in the blended-target domain, we propose a similarity-based style adaptation strategy for MBDA, which selects target styles to enhance the source representation space.

- We propose an uncertainty estimation technique to enhance the robustness of our method, which exploits valuable knowledge from multiple source domains. In addition, we construct a specific adversarial learning strategy for MBDA, which aligns domains without the requirement of domain labels.

## 2 Related Works

**Single-source and Single-target DA (SSDA).** The objective of SSDA is to learn domain-invariant representations through the relations between the source and target domains. Based on this objective, researchers have carried out widespread researches and achieved certain progress [17, 18, 19, 20, 21, 22, 23]. The current SSDA methods are mainly divided into two types: distance metric-based approaches [17, 18, 19] and adversarial learning-based approaches [20, 21, 22, 23]. Distance metric-based methods learn domain-invariant representations through feature discrepancy matching by using a distance metric function. DAN [18] utilizes multi-kernel maximum mean discrepancy (MMD) to measure the discrepancy between the source and target domains and then minimizes the discrepancy to learn domain-invariant representations. CAF [19] utilizes sliced Wasserstein distance (SWD) [24] to measure domain discrepancy. Motivated by the GANs [25, 26], adversarial learning-based SSDA methods have also been widely researched [20, 21, 22, 23]. Adversarial learning methods perform min-max two-player games between the category classifier and domain discriminator to learn domain-invariant representations. DANN [20], which was the earliest work in adversarial learning-based SSDA, successfully achieves domain-level adaptation via a gradient reverse layer. Different from DANN, SCDA [22] and DALN [23] remove the discriminator from their networks and model the adversarial relation between the feature extractor and category classifier. Although the above-mentioned approaches have achieved great success, due to the limitations of single source domain features and single target domain features, current SSDA methods still face some challenges in real applications.

**Multiple Domains DA.** The motivation of multiple domains DA is to explore more useful knowledge from multiple domains for the tasks. Many researchers have focused on multiple-domain DA and proposed many excellent methods [13, 14, 15, 8], including MSDA [14, 15, 27, 28], MTDA [8], BTDA [13, 12], and MMDA (multi-source and multi-target DA) [29]. M3SDA [14] utilizes moment matching to align domain distribution. DCA [15] further extracts the multiview features from multiple source domains and then utilizes multiple classifiers and pseudo-label learning strategy to align distributions. Meanwhile, in MTDA, CGCT [8] utilizes feature aggregation and curriculum learning to learn the pseudo-labels of multiple target domains. AMDA [29] constructs multiple domain discriminators and utilizes attention mechanism to address the MMDA issue.

Recently, a more realistic DA scenario, BTDA, has been studied [12, 13]. For example, MCDA [13] utilizes the mutual condition to learn domain-invariant representations, which has achieved great progress in BTDA. However, single source domain in BTDA cannot provide sufficient feature information for aligning the source and blended-target domains. Furthermore, the unseen domain labels of the target domains also aggravate the challenges. Therefore, we consider multiple source domains in BTDA and utilize the style information of the target domains to optimize the representations of source features and minimize the model uncertainty, thereby obtaining a better transfer.

## 3 Method

### 3.1 Preliminary

In MBDA, we have $M$ labeled source domains $\mathcal{S} = \{\mathcal{S}_m\}_{m=1}^M$ that are drawn from distributions $\{P_{\mathcal{S}_m}\}_{m=1}^M$. $\mathcal{S}_m = \{x_i^{\mathcal{S}_m}, y_i^{\mathcal{S}_m}\}_{i=1}^{|\mathcal{S}_m|}$, where $x_i^{\mathcal{S}_m} \in \mathbb{R}^d$ denotes the $i$-th source sample from the $m$-th source domain and $y_i^{\mathcal{S}_m}$ is the corresponding category label, and $d$ denotes the number of dimensions. Meanwhile, we have an unlabeled blended-target domain $\mathcal{T}$ that consists of $N$ sub-target domains $\{\mathcal{T}_n\}_{n=1}^N$, and $\mathcal{T} = \{x_j^{\mathcal{T}}\}_{j=1}^{|\mathcal{T}|}$. The distributions of sub-target domains are $\{P_{\mathcal{T}_n}\}_{n=1}^N$. Therefore, the distribution of blended-target domain $P_{\mathcal{T}}$ is the mixture of sub-target domains, *i.e.*, $P_{\mathcal{T}} = \sum_{n=1}^N \pi_n P_{\mathcal{T}_n}$, where $\pi \in [0,1]$ and $\sum_{n=1}^N \pi_n = 1$. Each of the source and target domains shares the same category space. The objective of MBDA is to train a model that utilizes multiple source domain features and performs well on the blended-target domain. Different from MTDA, the target domain labels are unseen in MBDA. In addition, the analysis in [12] demonstrated that directly utilizing DA methods to address BTDA transfer tasks may cause increased uncertainty and negative transfer. Therefore, we utilize the style information of the blended-target domain to augment source features and minimize the uncertainty of the model. Furthermore, the adversarial learning strategy in our method without the requirement of domain labels of the sub-target domains is suitable for MBDA setting. Figure 1 illustrates the overall architecture of SAUE.

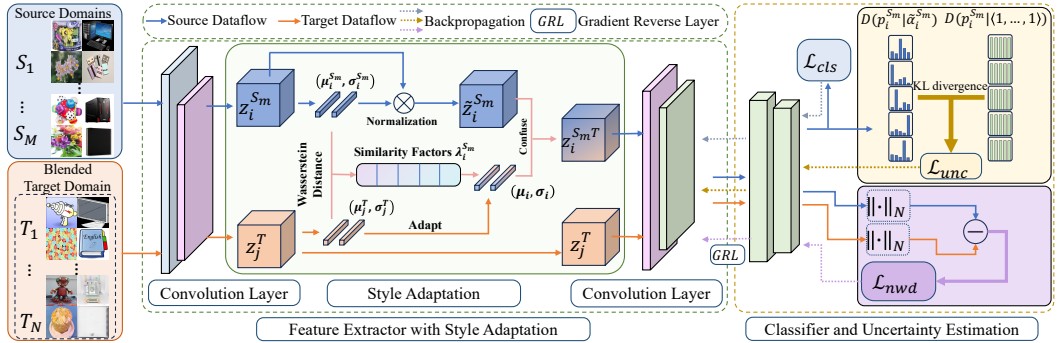

Figure 1: Overview of the proposed SAUE approach. First, the style information of the blended-target domain is utilized to augment the source features through the similarity factors. Second, we calculate the uncertainty of the model and optimize prediction uncertainty via the Dirichlet distribution. Finally, the adversarial learning strategy without discriminator effectively guides the SAUE process to adapt the blended-target domain without the requirement of domain labels of sub-target domains.

## 3.2 Style Adaptation from Blended-Target Domain

Since the principal parts of features from different domains remain domain-invariant, the domain-specific parts, which mainly contain style information, are the main discrepancies between different domains. In addition, the target feature distributions in MBDA are confused, which may cause model degradation. Therefore, we try to utilize the style information of blended-target domain to augment source features, which brings source features closer to target features. Previous work [13] has demonstrated that low-level features of deep neural networks (DNNs) mainly represent style information. Some works [13, 30] have utilized the channel-wise mean and variance of the low-level features to represent the style information of input samples. Thus, for sample $x_i^{\mathcal{S}_m}$ which from the $m$-th source domain, let its low-level feature be $z_i^{\mathcal{S}_m} \in \mathbb{R}^{d \times H_{\mathcal{S}_m} \times W_{\mathcal{S}_m}}$, where $d$ denotes the channel and $H_{\mathcal{S}_m}$, $W_{\mathcal{S}_m}$ denote the height and width of sample $x_i^{\mathcal{S}_m}$. The channel-wise mean and variance of the low-level feature $z_i^{\mathcal{S}_m}$ can be defined as follows:

$$\mu_i^{\mathcal{S}_m} = \frac{1}{H_{\mathcal{S}_m} W_{\mathcal{S}_m}} \sum_{h=1}^{H_{\mathcal{S}_m}} \sum_{w=1}^{W_{\mathcal{S}_m}} z_i^{\mathcal{S}_m}, \ \sigma_i^{\mathcal{S}_m} = \sqrt{\frac{1}{H_{\mathcal{S}_m} W_{\mathcal{S}_m}} \sum_{h=1}^{H_{\mathcal{S}_m}} \sum_{w=1}^{W_{\mathcal{S}_m}} (z_i^{\mathcal{S}_m} - \mu_i^{\mathcal{S}_m})^2}. \tag{1}$$

Low-level features mainly represent style information, but different samples contain specific pieces of style information. Therefore, we adopt feature normalization technique to standardize the feature $z_i^{\mathcal{S}_m}$, and the normalized feature $\tilde{z}_i^{\mathcal{S}_m}$ is defined as:

$$\tilde{z}_i^{\mathcal{S}_m} = \frac{z_i^{\mathcal{S}_m} - \mu_i^{\mathcal{S}_m}}{\sigma_i^{\mathcal{S}_m} + \epsilon}, \tag{2}$$

where $\epsilon$ is a small value used to avoid division by zero.

Then, the target features will be utilized to augment the normalized source features. Previous work [13] randomly augmented source features through target style information, which yielded limited performance. In our work, we select the target style information according to a weight factor. Specifically, we leverage the Wasserstein Distance [31] to measure the style distribution discrepancy $w_i^{\mathcal{S}_m}$ between the source low-level feature $z_i^{\mathcal{S}_m}$ and the target low-level feature $z_j^{\mathcal{T}}$ as follows:

$$w_i^{\mathcal{S}_m} = \|\mu_i^{\mathcal{S}_m} - \mu_j^{\mathcal{T}}\| + (\sigma_i^{\mathcal{S}_m})^2 + (\sigma_j^{\mathcal{T}})^2 - 2\sigma_i^{\mathcal{S}_m}\sigma_j^{\mathcal{T}}. \tag{3}$$

Then, we utilize $w_i^{\mathcal{S}_m}$ to calculate the weight factor as follows:

$$\lambda_i^{\mathcal{S}_m} = \frac{\exp(1/(1 + w_i^{\mathcal{S}_m}))}{\sum_{m=1}^{M} \sum_{i=1}^{|\mathcal{S}_m|} \exp(1/(1 + w_i^{\mathcal{S}_m}))}. \tag{4}$$

To ensure that the sum in Eq. (4) equals to 1, we utilize softmax operation to normalize each $\lambda_i^{\mathcal{S}_m}$. Then, we can obtain the weighted target style as follows:

$$\mu_i = \sum_{m=1}^{M} \sum_{i=1}^{|\mathcal{S}_m|} \lambda_i^{\mathcal{S}_m} \mu_j^{\mathcal{T}}, \ \sigma_i = \sum_{m=1}^{M} \sum_{i=1}^{|\mathcal{S}_m|} \lambda_i^{\mathcal{S}_m} \sigma_j^{\mathcal{T}}. \tag{5}$$

Finally, the low-level source feature augmented by the blended-target style information can be calculated as follows:

$$z_i^{\mathcal{S}_m \mathcal{T}} = \sigma_i \tilde{z}_i^{\mathcal{S}_m} + \mu_i. \tag{6}$$

Different from previously developed image augmentation method [13], our method directly utilizes target information with weight factor to augment source features instead of generating specific images. The low-level feature $z_i^{\mathcal{S}_m \mathcal{T}}$ augmented by diverse target styles further mitigates the impact of the domain-specific attributes.

### 3.3 Uncertainty Estimation and Elimination

Although multiple source domains provide additional supervised information for adaptation compared to a single source domain, they also introduce more domain-specific attributes. This can cause model degradation, especially when some source domains are significantly dissimilar to the blended-target domain due to the abundance of domain-specific attributes. Evidential model learning (EDL) [32] is an interpretable approach that fuses knowledge from multiple domains using the Dempster-Shafer Rule [33, 34], which is more beneficial to MBDA scenario. Thus, we utilize the Dirichlet-based evidential model [32] to estimate the uncertainty of our model during the training process. Specifically, for a sample $x_i$, we have the predictions $p_i = C(G(x_i)) = [p_{i1}, p_{i2}, \ldots, p_{iK}]$, where $C$ and $G$ denote the classifier and feature generator, respectively. The probability density function of $p_i$ is defined as follows:

$$D(p_i|\alpha_i) = \begin{cases} \frac{1}{B(\alpha_i)} \prod_{k=1}^{K} p_k^{\alpha_{ik}-1} & \text{for} \quad p \in U_K \\ 0 & \text{otherwise} \end{cases}, \tag{7}$$

where $U_k = \{p_i | \sum_{k=1}^{K} p_{ik} = 1$ and $0 \leq p_{i1}, ..., p_{iK} \leq 1\}$ is the $K$-dimensional unit simplex and $\alpha_i$ is the parameter of the Dirichlet distribution. $B(\alpha_i) = \frac{\Gamma(\sum_{k=1}^{K} \alpha_{ik})}{\prod_{k=1}^{K} \Gamma(\alpha_{ik})}$ is the $K$-dimensional multinomial beta function, and $\Gamma(\cdot)$ denotes the gamma function.

Previous work [32] has demonstrated that DNNs can generate opinions for classification tasks as Dirichlet distributions. Therefore, given sample $x_i$, for prediction of class $c$ that generated by DNNs, the Dirichlet distributions connected with DNNs can be defined as follows:

$$P(y = c|x_i) = \frac{\alpha_{ic}}{\sum_{k=1}^{K} \alpha_{ik}} = \frac{C_c(G(x_i))}{\sum_{k=1}^{K} C_k(G(x_i))} = \mathbb{E}[D(p_{ic}|\alpha_i)]. \tag{8}$$

The derivation of Eq. (8) is provided in Appendix B.

In this work, for source sample $x_i^{\mathcal{S}_m}$, we utilize the prediction of the category classifier as the evidence vector, and the parameters of the corresponding Dirichlet distribution can be defined as $\alpha_i^{\mathcal{S}_m} = C(G(x_i^{\mathcal{S}_m})) + 1$. To eliminate the uncertainty, we force the total evidence to zero when the model generates an incorrect prediction for the source sample, and the corresponding uniform Dirichlet distribution is $D(p_i^{\mathcal{S}_m}|\langle 1, \ldots, 1 \rangle)$. For implementation, we utilize the Kullback-Leibler (KL) divergence to reduce the impact of incorrectly classified source samples in our loss function, which is defined as follows:

$$\mathcal{L}_{unc}(x^{\mathcal{S}_m}) = \sum_{i=1}^{|\mathcal{S}_m|} KL[D(p_i^{\mathcal{S}_m}|\tilde{\alpha}_i^{\mathcal{S}_m}) \| D(p_i^{\mathcal{S}_m}|\langle 1, \ldots, 1 \rangle)], \tag{9}$$

where $\tilde{\alpha}_i^{\mathcal{S}_m} = y_i^{\mathcal{S}_m} + (1 - y_i^{\mathcal{S}_m}) \odot \alpha_i^{\mathcal{S}_m}$ denotes the Dirichlet parameter used to remove the true evidence from prediction $\alpha_i^{\mathcal{S}_m}$. Specifically, the KL divergence is:

$$KL[D(p_i^{\mathcal{S}_m}|\tilde{\alpha}_i^{\mathcal{S}_m}) \| D(p_i^{\mathcal{S}_m}|\langle 1, \ldots, 1 \rangle)]$$
$$= \log \left[ \frac{\Gamma(\sum_{k=1}^{K} \tilde{\alpha}_i^{\mathcal{S}_m})}{\Gamma(K) \prod_{k=1}^{K} \Gamma(\tilde{\alpha}_{ik})} \right] + \sum_{k=1}^{K} (\tilde{\alpha}_{ik} - 1) \left[ \Psi(\tilde{\alpha}_{ik}) - \Psi(\sum_{j=1}^{K} \tilde{\alpha}_{ij}) \right], \tag{10}$$

where $\Gamma(\cdot)$ and $\Psi(\cdot)$ denotes the gamma function and digamma function, respectively.

---

**Algorithm 1** SAUE for MBDA

---

**Input:** Source domains $\{\mathcal{S}_m\}_{m=1}^M$ and the corresponding data $\{x_i^{\mathcal{S}_m}, y_i^{\mathcal{S}_m}\}_i^{|\mathcal{S}_m|}$, blended-target domain data $\{x_j^{\mathcal{T}}\}$, hyper-parameters $\lambda_e$ and $\lambda_d$, maximum iteration $I$, and mini-batch size $B$.

**Output:** Optimal feature generator $G$ and category classifier $C$.

1: **for** $i$ in 1:$I$ **do**
2:      Randomly sample a mini-batch of $B$ source samples and target samples.
3:      Augment the source features by utilizing style adaptation, *i.e.*, Eq. (6).
4:      Minimize the parameters of the feature generator and category classifier by $\mathcal{L}_{cls}$.
5:      Optimize the uncertainty of model through $\mathcal{L}_{unc}$.
6:      Perform the min-max game between feature generator and classifier with $\mathcal{L}_d$:

$$\min_G \max_C \mathcal{L}_d(x^{\mathcal{S}_m}, x^{\mathcal{T}}).$$

---

## 3.4 Domain Adversarial Alignment without Domain Labels

Existing works [20, 21, 22, 8] have demonstrated that adversarial learning strategy is beneficial in DA. However, most of the adversarial learning strategies in DA [20, 21] usually request the domain labels of the source and target domains to train their discriminators, which cannot satisfy MBDA. Inspired by the intra- and inter-class correlation [35], we design an adversarial learning strategy for MBDA without discriminator and domain label requirements. Specifically, the category classifier is reused to discriminate which domain a feature originates from, with the guidance of the Nuclear norm $\|\cdot\|_*$. We first measure the distribution difference between the source and blended-target domains through the nuclear-norm 1-Wasserstein discrepancy (NWD) [23] and then utilize a gradient reverse layer (GRL) [20] to maximize the discriminative loss of the classifier. Simultaneously, we minimize the feature generator to play the min-max game with the classifier through the NWD. First, the NWD loss can be defined as:

$$\mathcal{L}_d(x^{\mathcal{S}_m}, x^{\mathcal{T}}) = \frac{1}{|\mathcal{S}_m|} \sum_{i=1}^{|\mathcal{S}_m|} \left\| C(G(x_i^{\mathcal{S}_m})) \right\|_* - \frac{1}{|\mathcal{T}|} \sum_{j=1}^{|\mathcal{T}|} \left\| C(G(x_j^{\mathcal{T}})) \right\|_*. \tag{11}$$

Then, the adversarial learning strategy between feature generator and classifier is defined as follows:

$$\min_G \max_C \mathcal{L}_d(x^{\mathcal{S}_m}, x^{\mathcal{T}}). \tag{12}$$

## 3.5 Model Optimization and Theoretical Analysis

**Overall Objective.** The overall loss function that optimizes SAUE for MBDA is defined as:

$$\min_{G,C} \left\{ \mathcal{L}_{cls}(x^{\mathcal{S}_m}, x^{\mathcal{T}}) + \lambda_e \mathcal{L}_{unc}(x^{\mathcal{S}_m}) + \lambda_d \max_C \mathcal{L}_d(x^{\mathcal{S}_m}, x^{\mathcal{T}}) \right\}, \tag{13}$$

where $\lambda_e = \min(1, e/\lambda_e') \in [0, 1]$ is the annealing coefficient, which prevents $\mathcal{L}_{unc}$ from over-penalizing the neural network to a uniform distribution in the early training epochs, and $e$ is the current number of epochs and $\lambda_e' = 40$. $\lambda_d$ is a hyper-parameter which is initially set to $\lambda_d = 1$ as in [23]. $\mathcal{L}_{cls}$ is the classification loss, which ensures that the category classifier can correctly classify samples. With the the cross-entropy loss $\mathcal{L}_{ce}$, the classification loss $\mathcal{L}_{cls}$ is defined as follows:

$$\mathcal{L}_{cls}(x^{\mathcal{S}_m}, y^{\mathcal{S}_m}) = \sum_{m=1}^M \frac{1}{|\mathcal{S}_m|} \sum_{i=1}^{|\mathcal{S}_m|} \mathcal{L}_{ce}(C(G(x_i^{\mathcal{S}_m})), y_i^{\mathcal{S}_m}). \tag{14}$$

After adversarial training, our model can effectively adapt the blended-target domain without the requirement of domain labels of sub-target domains. The concrete steps of SAUE are shown in Algorithm 1.

**Theoretical Analysis.** Here, we utilize PAC-Bayesian theory [36] to optimize our classification model with uncertainty estimation and elimination. The full-bound theorem motivated by previous work [37] is illustrated in Theorem 1 and Lemma 1, and the proofs are provided in Appendix C.

**Theorem 1** [37]. *Suppose we have given the m-th source data distribution $P_{\mathcal{S}_m}$, a hypothesis set $\mathcal{H}$, and a prior distribution $\pi$ over the hypothesis space $\Theta$. For any $\tau \in (0,1]$ and $\lambda > 0$, with a probability at least $1 - \tau$ over the source samples $\mathcal{S}_m \sim P_{\mathcal{S}_m}$, for all posteriors $\rho$, we have:*

$$\mathbb{E}_{\rho(\mathcal{H})}[\mathcal{L}(\mathcal{H})] \leq \mathbb{E}_{\rho(\mathcal{H})}[\tilde{\mathcal{L}}_{\mathcal{S}_m}(\mathcal{H})] + \frac{1}{\lambda}\left[KL(\rho\|\pi) + \log\frac{1}{\tau} + \Psi_{\mathcal{S}_m,\pi}(\lambda,n)\right], \quad (15)$$

*where $\Psi_{\mathcal{S}_m,\pi}(\lambda,n) = \log\mathbb{E}_{\pi(\mathcal{H})}\mathbb{E}_{\mathcal{S}_m \sim P_{\mathcal{S}_m}}\left[e^{\lambda(\mathcal{L}(\mathcal{H})-\mathcal{L}(\tilde{\mathcal{H}}))}\right].$*

**Lemma 1** [38]. *The PAC-Bayes bound, involving constants $\tau$ and $n$, as introduced in Theorem 1, is minimized by the Bayesian posterior $p(\mathcal{H})$, which represents the distribution over $\Theta$.*

During uncertainty estimation and elimination, just as in Eq. (9), we utilize $D(p_i^{\mathcal{S}_m}|\tilde{\alpha}_i^{\mathcal{S}_m})$ as the posterior distribution and $D(p_i^{\mathcal{S}_m}|\langle 1,\ldots,1\rangle)$ as the prior distribution. Therefore, the upper bound of the classification model can be expressed as:

$$\sum_{m=1}^{M}\frac{1}{|\mathcal{S}_m|}\sum_{i=1}^{|\mathcal{S}_m|}\left[\mathcal{L}_{cls} + \frac{1}{\lambda}KL(D(p_i^{\mathcal{S}_m}|\tilde{\alpha}_i^{\mathcal{S}_m})\|D(p_i^{\mathcal{S}_m}|\langle 1,...,1\rangle))\right]. \quad (16)$$

**Generalization Bound.** In this part, we prove why SAUE performs well on the blended-target domain via Lemma 2 and Theorem 2. The proofs and derivations are provided in Appendix D.

**Lemma 2** [39]. *Suppose we have given the probability measures $\nu_{\mathcal{S}_m}, \nu_{\mathcal{T}} \in \mathcal{P}(\mathcal{F})$ of the m-th source feature $f_{\mathcal{S}_m}$ and the blended-target domain feature $f_{\mathcal{T}}$, a hypothesis space $\Theta$, and a subspace $\tilde{\mathcal{H}} \in \Theta$. Let $\mathcal{F}$ denote a fixed representation space and $c(f_{\mathcal{S}_m}, f_{\mathcal{T}})$ denote the adaptation cost. For the ideal classifier $h' \in \tilde{\mathcal{H}}$ and any classifier $h \in \tilde{\mathcal{H}}$ with $f_{\mathcal{S}_m} \sim \nu_{\mathcal{S}_m}$ and $f_{\mathcal{T}} \sim \nu_{\mathcal{T}}$, we have:*

$$|\epsilon_{\mathcal{S}_m}(h,h') - \epsilon_{\mathcal{T}}(h,h')| \leq \frac{1}{2}d_{\mathcal{H}\Delta\mathcal{H}}(\nu_{\mathcal{S}_m}, \nu_{\mathcal{T}}), \quad (17)$$

*where $\epsilon_{\mathcal{S}_m}$ and $\epsilon_{\mathcal{T}}$ denote the error on the m-th source domain and the error on the blended-target domain respectively, and $\epsilon_{\mathcal{T}} = \frac{1}{N}\sum_{j=1}^{N}\epsilon_{\mathcal{T}_j}$. $d_{\mathcal{H}\Delta\mathcal{H}}$ denotes the $\mathcal{H}\Delta\mathcal{H}$-distance.*

**Theorem 2.** *Based on Lemma 2, with the error of the ideal joint hypothesis $\eta' = \epsilon_{\mathcal{S}_m}(h') + \epsilon_{\mathcal{T}}(h')$ which is a sufficiently small constant, for any $\delta \in (0,1)$, with probility at least $1-\delta$, for every $h \in \mathcal{H}$, $\epsilon_{\mathcal{T}}(h)$ is bounded by the following terms:*

$$\epsilon_{\mathcal{T}}(h) \leq \epsilon_{\mathcal{S}_m}(h) + \frac{1}{2}\hat{d}_{\mathcal{H}\Delta\mathcal{H}}(\nu_{\mathcal{S}_m}, \nu_{\mathcal{T}}) + 4\sqrt{\frac{2d\log(2b') + \log(\frac{2}{\delta})}{b'}} + \eta', \quad (18)$$

*where $\eta' = \epsilon_{\mathcal{S}_m}(h') + \epsilon_{\mathcal{T}}(h')$ is the ideal error for the classifier, which is a sufficiently small constant. $b'$ is the size of unlabeled samples.*

Therefore, the final objective of the MBDA classification task is to reduce the joint domain discrepancy term $\sum_{m=1}^{M}|\epsilon_{\mathcal{S}_m}(h,h') - \epsilon_{\mathcal{T}}(h,h')|$.

## 4 Experiments

### 4.1 Datasets and Implementation Details

**Datasets.** Four standard benchmark datasets are used to validate the effectiveness of our proposed method. The **ImageCLEF-DA** [40] contains 2,400 images and is divided into 4 domains: Bing (b), Caltech (c), ImageNet (i), and Pascal (p). Each domain has 12 categories, and every category has 50 images. The **Office-Home** [41] also consists of 4 domains and 15,588 images belonging to 65 categories from four subdomains: Art (Ar), Clipart (Cl), Products (Pr), and Real world (Rw). The **DomainNet** [14] is a large-scale dataset in DA that contains 0.6 million images of 345 categories from 6 domains: Clipart (C), Infograph (I), Painting (P), Quickdraw (Q), Real world (R), and Sketch (S). Following the protocol used in [29], we select 126 categories and 4 domains (C, P, R, and S) in our experiments. The **VisDA 2017** [42] dataset is a challenging dataset consists of 2 domains (Syn. and Rel.) and 7 categories.

**Implementation Details.** We utilize PyTorch framework [43] to perform our experiments; the PyTorch version is 1.13.1 and CUDA version is 11.7. We use an ImageNet pre-trained ResNet [44], replacing the last FC layer with task-specific FC layers. All experiments are run on a single GeForce RTX-4090 GPU, and the batch size of both the source and blended-target domains are set to 32. The optimizer is Stochastic Gradient Descent (SGD) with a momentum parameter of 0.9 and a weight decay of 1e-3. The learning rate is 1e-3 and updated by the LambdaLR [43] during the training process.

## 4.2 Comparisons to State-of-the-Art

To evaluate the effectiveness of our proposed method, we conduct extensive experiments and compare our approach with the state-of-the-art (SOTA) methods in terms of DA classification. The comparison methods include SSDA approaches, *i.e.,* MCD [45], DAN [18], TSA [46], DALN [23], BIWAA [47], and SCDA [22]; MSDA methods, *i.e.,* MDAN [48], DCTN [49], and DIDA [50]. MTDA/BTDA methods: MTDA-ITA [7] and MCDA [13]; and Multi-source Multi-target DA (MMDA), *i.e.,* AMDA [29] and HTA [51]. The comparison results are presented in Tables 2-4, in which we select two domains as source domains and combine other two domains to form the blended-target domain. Note that these approaches do not totally match the MBDA setting. Therefore, we utilize the following rule for our comparison. For SSDA setting, one column denotes one SSDA task, such as R→C in Table 2. For MSDA methods that contain more than two source domains, we implement those methods according to their released codes, reset the source domain into two domains, such as R+S→C, and mark them with "*". Similarly, under the MTDA and BTDA settings, we reset the target into two domains and select the highest one in MTDA/BTDA task group that contains the same target domains, such as R→ C+P and S→ C+P in Table 2. For MMDA setting, two domains are sources, and the other domains are targets, such as R+S→C+P in Table 2. For better comparison, all results in Tables 2-4 are the averages of two target domains.

**Results on the DomainNet** are displayed in Table 2. Our SAUE method achieves SOTA performance in most of the experimental groups and attains the best performance in terms of average accuracy. Compared to the BTDA method MCDA in multi-source setting, our method achieves better performance because the style information of the target domain selected by the weight factor can enhance the source feature representations. Compared to MMDA method AMDA, although AMDA can access the domain labels of the target domains, our method still overpasses

Table 2: Accuracy (%) on the DomainNet for MBDA (ResNet-50).

| Method | R+S / C+P | S+P / C+R | P+R / C+S | C+S / P+R | R+C / P+S | C+P / R+S | Avg. |
|---|---|---|---|---|---|---|---|
| DANN[20] JMLR'16 | 31.4 | 39.7 | 26.8 | 29.3 | 31.3 | 31.2 | 31.6 |
| DAN[18] TPAMI'19 | 32.8 | 40.6 | 28.2 | 29.8 | 31.5 | 32.0 | 32.5 |
| MDAN[48] NeurIPS'18 | 54.5 | 59.0 | 45.0 | 58.8 | 51.7 | 61.0 | 54.5 |
| MTDA[7] TIP'20 | 52.4 | 48.7 | 45.5 | 53.3 | 51.5 | 52.0 | 50.5 |
| AMDA[29] TIP'21 | 65.8 | 67.8 | 56.7 | 65.1 | 58.9 | 66.4 | 63.4 |
| DALN*[23] CVPR'22 | 61.2 | 69.2 | 64.1 | 63.5 | 59.3 | 64.8 | 63.7 |
| MCDA*[13] AAAI'23 | 62.2 | 68.7 | 61.7 | 63.4 | 61.2 | 65.4 | 63.8 |
| DGWA*[52] TMM'24 | 66.4 | 71.3 | 63.4 | 67.5 | 64.6 | 70.2 | 65.0 |
| **SAUE (Ours)** | **70.8** | **76.9** | **67.6** | **71.9** | **65.2** | **73.1** | **70.9** |

"*" denotes that the results are obtained by the released code of the corresponding method. The best results are bolded.

AMDA in terms of average classification accuracy (overpass **7.5%**) and without the requirement of the domain labels of the target domains. Furthermore, both AMDA and our method are adversarial learning methods, and the comparison results further demonstrate the effectiveness of our adversarial learning strategy. These obtained improvements are mainly due to the uncertainty optimization process and the style information derived from target features.

Table 3: Accuracy (%) on the (a) Office-Home and the (b) ImageCLEF-DA for MBDA (ResNet-50).

(a) **Office-Home**

| Method | Rw+Pr / Ar+Cl | Cl+Rw / Ar+Pr | Pr+Cl / Ar+Rw | Rw+Ar / Cl+Pr | Ar+Pr / Cl+Rw | Cl+Ar / Pr+Rw | Avg. |
|---|---|---|---|---|---|---|---|
| DANN[20] JMLR'16 | 53.5 | 61.9 | 53.5 | 55.6 | 57.1 | 60.1 | 57.6 |
| DAN[18] TPAMI'19 | 53.4 | 60.1 | 52.2 | 54.3 | 52.2 | 58.7 | 56.3 |
| MTDA[7] TIP'20 | 51.9 | 64.9 | 60.3 | 59.4 | 58.2 | 62.4 | 59.5 |
| MDAN[48] NeurIPS'18 | 55.4 | 69.1 | 61.2 | 61.5 | 55.9 | 70.4 | 62.2 |
| SCDA[22] CVPR'21 | 64.1 | 74.7 | 70.0 | 68.3 | 68.7 | 77.6 | 70.1 |
| AMDA[29] TIP'21 | 61.4 | 77.0 | 72.3 | 67.4 | 64.9 | 77.4 | 70.0 |
| HTA[51] Appl. Intell.'23 | 62.2 | 78.9 | 75.0 | 68.7 | 66.2 | 79.0 | 71.9 |
| MCDA*[13] AAAI'23 | 63.6 | 74.9 | 70.0 | 68.7 | 68.1 | 78.1 | 70.6 |
| DGWA[52] TMM'24 | 63.7 | 78.6 | 73.9 | 70.7 | 66.9 | 78.8 | 72.1 |
| **SAUE (Ours)** | **65.6** | **79.9** | **75.2** | **70.1** | **71.8** | **79.3** | **73.7** |

(b) **ImageCLEF-DA**

| Method | i+p / b+c | p+c / b+i | c+i / b+p | b+p / c+i | i+b / c+p | b+c / i+p | Avg. |
|---|---|---|---|---|---|---|---|
| DANN[20] JMLR'16 | 76.4 | 72.4 | 69.1 | 87.9 | 82.9 | 79.3 | 77.9 |
| DAN[18] TPAMI'19 | 78.3 | 74.8 | 70.3 | 91.5 | 85.0 | 78.8 | 79.8 |
| CDAN*[21] NeurIPS'18 | 78.3 | 76.8 | 68.2 | 92.8 | 85.9 | 82.3 | 80.6 |
| AMDA[29] TIP'21 | 78.8 | 77.3 | 71.7 | 92.3 | 85.2 | 83.8 | 81.5 |
| SCDA*[22] CVPR'21 | 78.9 | 77.2 | 71.8 | 93.9 | 85.5 | 85.0 | 82.1 |
| DIDA*[50] TIP'22 | 78.9 | 77.9 | 72.0 | 92.2 | 86.8 | 85.1 | 82.2 |
| HTA[51] Appl. Intell.'23 | 79.3 | 78.2 | 72.3 | 92.8 | 85.6 | 84.9 | 82.2 |
| MCDA*[13] AAAI'23 | 77.4 | 79.1 | 70.3 | 91.8 | 86.2 | 83.8 | 81.4 |
| DGWA[52] TMM'24 | 79.7 | 79.1 | 72.7 | 93.8 | 86.0 | 84.5 | 82.7 |
| **SAUE (Ours)** | **80.8** | **80.2** | **73.8** | **94.9** | **88.6** | **87.2** | **84.3** |

"*" denotes that the results are obtained by the released code of the corresponding method. The best results are bolded.

**Results on the Office-Home** are shown in Table 3a. The experimental results are compared with those of the SOTA methods, illustrating that our proposed method achieves dramatic improvements in most comparison groups and achieves the highest average classification accuracy (**73.7%**). Note that the Rw domain contains a total of 34,856 images, which is far more numerous than the other

Table 4: Accuracy (%) on the (a) default version of DomainNet dataset and (b) VisDA-2017 dataset (ResNet-101).

(a) DomainNet

| Method | C+P+Q / R+S | C+P+R / Q+S | C+P+S / Q+R | C+Q+R / P+S | C+Q+S / P+R | C+R+S / P+Q | P+Q+R / C+S | P+Q+S / C+R | P+R+S / C+Q | Q+R+S / C+P | Avg. |
|---|---|---|---|---|---|---|---|---|---|---|---|
| MCDA | 54.6 | 28.5 | 39.1 | 50.3 | 52.9 | 30.6 | 53.2 | 60.1 | 34.2 | 56.2 | 46.1 |
| SCDA | 54.2 | 31.0 | 40.5 | 51.8 | 56.2 | 30.4 | 54.9 | 59.3 | 36.0 | 57.0 | 47.1 |
| DGWA | 54.7 | 31.3 | 40.7 | 52.2 | 56.8 | 30.7 | 55.3 | 59.6 | 36.2 | 57.5 | 47.4 |
| **SAUE (Ours)** | **57.7** | **34.6** | **42.9** | **54.7** | **59.2** | **36.9** | **56.0** | **63.6** | **37.1** | **58.7** | **50.2** |

| Method | R+S / C+P+Q | Q+S / C+P+R | Q+R / C+P+S | P+S / C+Q+R | P+R / C+Q+S | P+Q / C+R+S | C+S / P+Q+R | C+R / P+Q+S | C+Q / P+R+S | C+P / Q+R+S | Avg. |
|---|---|---|---|---|---|---|---|---|---|---|---|
| MCDA | 40.0 | 54.6 | 50.2 | 33.4 | 41.2 | 52.9 | 45.3 | 40.2 | 48.3 | 41.2 | 44.7 |
| SCDA | 41.2 | 54.7 | 50.2 | 33.8 | 41.3 | 54.2 | 46.5 | 39.3 | 49.4 | 42.1 | 45.4 |
| DGWA | 41.5 | 55.1 | 50.7 | 34.3 | 42.6 | 54.7 | 46.9 | 39.7 | 49.8 | 42.6 | 45.8 |
| **SAUE (Ours)** | **43.3** | **57.7** | **53.1** | **37.7** | **44.6** | **57.5** | **50.5** | **41.2** | **51.3** | **45.3** | **48.2** |

(b) VisDA-2017

| Method | Syn.→Rel. |
|---|---|
| MCD | 71.9 |
| SWD | 76.4 |
| BNM | 70.4 |
| TSA | 78.6 |
| SCDA | 79.7 |
| DALN | 80.6 |
| DGWA | 80.3 |
| **SAUE(ours)** | **81.5** |

three domains. Therefore, the adaptation task faces larger domain shifts and extremely unbalanced classes. The proposed method still achieves 2.8% improvements over AMDA and achieves dramatic improvements in the Rw+Pr→Ar+Cl and Rw+Ar→Pr+Cl tasks. These results occur because the proposed method decreases the impact of unbalanced classes by enhancing the feature representations and optimizing the prediction uncertainty.

**Results on the ImageCLEF-DA** are provided in Table 3b. Compared with the SOTA methods, our proposed method achieves an average accuracy of **84.3%**, outperforming the existing approaches. Note that all four domains in ImageCLEF-DA contain 600 images. Therefore, the experimental results further demonstrate that our proposed method is effective when all the domains contain the same samples and classes.

**Results on the Default Version of DomainNet and VisDA 2017**. To evaluate the effectiveness of SAUE in different numbers of the source and target domains. We perform comparisons on the default version of the DomainNet dataset and the VisDA 2017 dataset, respectively. The default version of the DomainNet dataset consists of 5 domains, leading to the division of transfer tasks for MBDA into two categories: C+P+Q→R+S and R+S→C+P+Q. As shown in Table 4, SAUE outperforms the comparison methods across all transfer tasks, achieving the highest average classification accuracy. These results from large-scale datasets further demonstrate the superiority and flexibility of SAUE.

## 4.3 Experiment Analysis

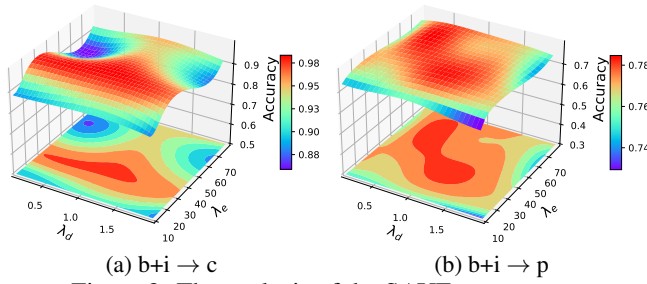

(a) b+i → c          (b) b+i → p

Figure 2: The analysis of the SAUE parameters.

| Choices | $\lambda_d$ =0.1 | $\lambda_d$ =0.5 | $\lambda_d$ =1.0 | $\lambda_d$ =1.5 | $\lambda_d$ =2.0 | Choices | $\lambda_d$ =0.1 | $\lambda_d$ =0.5 | $\lambda_d$ =1.0 | $\lambda_d$ =1.5 | $\lambda_d$ =2.0 |
|---|---|---|---|---|---|---|---|---|---|---|---|
| $\lambda_e$ =10 | 96.5 | 96.7 | 96.8 | 96.6 | 96.5 | $\lambda_e$ =10 | 77.5 | 77.6 | 78.1 | 77.9 | 77.4 |
| $\lambda_e$ =20 | 96.8 | 97.0 | 97.1 | 97.0 | 96.9 | $\lambda_e$ =20 | 77.7 | 78.1 | 78.2 | 78.0 | 77.9 |
| $\lambda_e$ =40 | 97.0 | 97.2 | 97.3 | 97.2 | 97.2 | $\lambda_e$ =40 | 78.2 | 78.3 | 78.3 | 78.2 | 78.1 |
| $\lambda_e$ =60 | 94.7 | 94.9 | 95.2 | 95.1 | 94.9 | $\lambda_e$ =60 | 77.7 | 77.9 | 78.0 | 77.8 | 75.9 |
| $\lambda_e$ =80 | 88.7 | 93.3 | 93.4 | 93.2 | 89.8 | $\lambda_e$ =80 | 76.2 | 77.3 | 77.9 | 76.2 | 74.8 |

(a) b+i → c          (b) b+i → p

Table 5: The detailed numerical results corresponding to the relevant tasks.

**Sensitivity Analysis.** We evaluate the model's performance under different hyperparameter choices. Note that the hyperparameters in our method are the adversarial learning balance parameter $\lambda_d$ and annealing parameter $\lambda_e$. As shown in Figure 2, we test different parameter groups to analyze the parameter sensitivity of our method, where $\lambda_d = \{0.1, 0.5, 1.0, 1.5, 2.0\}$ and $\lambda_e = \{10, 20, 40, 60, 80\}$. The corresponding numerical results of Figure 2 are illustrated in Table 5. In Figure 2, our method is not sensitive to $\lambda_d$ and the best parameter choice is $\lambda_d = 1.0$, but very sensitive to $\lambda_e$ (with $\lambda_e = 40$

working best); if $\lambda_e$ too small, the model will suffer from tremendous degradation due to the model is over-penalized by uncertainty loss.

**Ablation Study.** As listed in Table 6, we conduct ablation experiments to demonstrate the effectiveness of the style adaptation module and the loss function of the uncertainty optimization process. We test three experimental groups in the DomainNet dataset, including that: 1) remove the style adaptation (SA) module, 2) remove the uncertainty loss $\mathcal{L}_{unc}$, and 3) remove both of the above-mentioned items.

Table 6: Ablation study of SAUE on the Domain-Net.

| Source Target | R+S C+P | S+P C+R | P+R C+S | C+S P+R | R+C P+S | C+P R+S | Avg. |
|---|---|---|---|---|---|---|---|
| w/o SA | 68.4 | 72.1 | 64.2 | 70.0 | 60.8 | 70.1 | 67.6 |
| w/o $\mathcal{L}_{unc}$ | 69.7 | 75.0 | 65.8 | 71.2 | 64.8 | 72.6 | 69.9 |
| w/o both | 65.1 | 71.9 | 63.8 | 68.8 | 60.2 | 69.3 | 66.5 |
| w/o WD | 70.2 | 77.0 | 66.3 | 71.6 | 64.1 | 71.5 | 70.1 |
| **SAUE** | **70.8** | **76.9** | **67.6** | **71.9** | **65.2** | **73.1** | **70.9** |

The results illustrate that both style adaptation and prediction uncertainty optimization are useful for MBDA. Furthermore, we explore different style adaptation techniques. We directly change the Wasserstein Distance (WD) to randomly selected style information and report the obtained results in the fourth row of middle part of Table 6. Compared with the random augmentation methods, our proposed method can better select the style information through weight factors, which enhances the feature representations of the source domains and reduces the impact of domain shifts. More experiment analysis is provided in Appendix E.

Table 7: Comparison about the SA module on the DomainNet dataset with different backbones.

| Setting | C+P+Q R+S | C+P+R Q+S | C+P+S Q+R | C+Q+R P+S | C+Q+S P+R | C+R+S P+Q | P+Q+R C+S | P+Q+S C+R | P+R+S C+Q | Q+R+S C+P | Avg. |
|---|---|---|---|---|---|---|---|---|---|---|---|
| without SA (ResNet-50) | 51.2 | 28.6 | 36.7 | 49.6 | 52.9 | 30.2 | 51.0 | 57.8 | 30.3 | 51.1 | 43.9 |
| with SA (ResNet-50) | 53.9 | 30.7 | 39.8 | 51.3 | 55.7 | 33.7 | 52.8 | 60.4 | 33.7 | 55.3 | 46.7 (**+2.8**) |
| without SA (ResNet-101) | 56.1 | 33.7 | 41.3 | 53.1 | 57.3 | 35.1 | 54.8 | 61.2 | 36.4 | 56.7 | 48.6 |
| with SA (ResNet-101) | 57.7 | 34.6 | 42.9 | 54.7 | 59.2 | 36.9 | 56.0 | 63.6 | 37.1 | 58.7 | 50.2 (**+1.6**) |

| Setting | R+S C+P+Q | Q+S C+P+R | Q+R C+P+S | P+S C+Q+R | P+R C+Q+S | P+Q C+R+S | C+S P+Q+R | C+R P+Q+S | C+Q P+R+S | C+P Q+R+S | Avg. |
|---|---|---|---|---|---|---|---|---|---|---|---|
| without SA (ResNet-50) | 39.5 | 53.2 | 47.3 | 33.2 | 40.1 | 52.4 | 45.8 | 38.2 | 45.3 | 40.5 | 43.5 |
| with SA (ResNet-50) | 41.1 | 55.3 | 49.8 | 35.6 | 42.5 | 54.7 | 48.3 | 39.5 | 47.7 | 43.2 | 45.8 (**+2.3**) |
| without SA (ResNet-101) | 42.1 | 56.5 | 51.6 | 35.2 | 43.8 | 57.0 | 49.4 | 39.9 | 50.7 | 43.1 | 46.9 |
| with SA (ResNet-101) | 43.3 | 57.7 | 53.1 | 37.7 | 44.6 | 57.5 | 50.5 | 41.2 | 51.3 | 45.3 | 48.2 (**+1.3**) |

**Effectiveness of the SA Module with Different Backbones.** We have analyzed the performance of our method using the ResNet-50 and ResNet-101 backbones in the default version of the DomainNet dataset in Table 7. The experimental results show that our method achieves significant performance gains when utilizing powerful backbones (ResNet-101). We compared the performance gains of the style adaptation module with standard backbone (ResNet-50) and powerful backbone (ResNet-101). The results indicate that the performance gains (+2.8% and +2.3%) of the style adaptation module when integrated into the ResNet-50 backbone surpass those (+1.6% and +1.3%) when integrated into the ResNet-101 backbone.

**Effectiveness of the SAUE with ResNet and ViT backbones.** We further evaluate the model's performance using different backbones, as shown in Table 8. When using ViT as the backbone, SAUE outperforms its performance with the ResNet-50 backbone. This

Table 8: Comparison about different backbones on the Office-Home dataset.

| Method | Rw+Pr Ar+Cl | Cl+Rw Ar+Pr | Pr+Cl Ar+Rw | Rw+Ar Cl+Pr | Ar+Pr Cl+Rw | Cl+Ar Pr+Rw | Avg. |
|---|---|---|---|---|---|---|---|
| ResNet-50 | 65.6 | 79.9 | 75.2 | 70.1 | 71.8 | 79.3 | 73.7 |
| ViT-B/16 | 69.2 | 83.1 | 79.1 | 73.6 | 74.5 | 83.7 | 77.2 |

performance gain is primarily due to the larger number of tunable parameters in ViT-B/16, demonstrating that our method effectively leverages these additional parameters to exploit transferable knowledge from multiple domains.

## 5    Conclusion

In this paper, we propose a SAUE approach for MBDA, which utilizes information from multiple source domains to adapt a blended-target domain. In particular, the style adaptation process utilizes similarity factors to select target style information to enhance the representations of the source features. The uncertainty estimation procedure utilizes the Dirichlet distribution to estimate the uncertainty of the model and then adopts the KL divergence measure to optimize the prediction uncertainty. The discriminator-free adversarial learning strategy is beneficial for MBDA. Extensive experimental results demonstrate the superior performance of SAUE to that of the competing methods.

## Acknowledgment

This work was supported by the National Natural Science Foundation of China (Grant No. 62176162) and the Guangdong Basic and Applied Basic Research Foundation (2023A1515012875, 2022A1515140099).

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

## Appendix Contents

This supplementary material provides more details that are not presented in the main paper due to space limitations. The organization is as follows:

## A  Broader Impacts & Limitations

D provides the proof and derivations of the generalization bound.

Our work focuses on the problem of multi-source blended-target domain adaptation (MBDA), which aims to train a model that generalizes well on an unlabeled and distribution-confused target domain by leveraging multiple labeled source domains. The effectiveness of our method on several real-world datasets suggests that it can benefit relevant applications and communities dealing with domain shifts, such as encrypted data analysis, medical imaging, and autonomous driving. Nevertheless, we should also be cautious about potential failures of our method when encountering more significant distribution shifts, such as an increased number of source or target domains. In the future, we plan to incorporate more sub-domains into our experiments for further verifying the performance of our method.

## B  Derivations of $P(y = c|x_i) = \mathbb{E}[D(p_{ic}|\alpha_i)]$

Given sample $x_i$, for prediction of class c that generated by DNNs can be calculated as:

$$
\begin{aligned}
&P(y = c|x_i) \\
&= \int \rho(y = c|p_i)\rho(p_i|x_i)dp_i \\
&= \int p_{ic} \cdot \rho(p_i|x_i)dp_i \\
&= \int \int \cdots \int p_{ic} \cdot \rho(p_{i1}, p_{i2,\ldots,p_{iK}}|x_i)dp_{i1}dp_{i2}\cdots dp_{iK} \\
&= \int p_{ic} \cdot \rho(p_{ic}|x_i)dp_{ic},
\end{aligned}
\tag{1}
$$

where $p_i = C(G(x_i)) = [p_{i1}, p_{i2}, \ldots, p_{iK}]$ and $p_{ic}$ is the $c$-th element of $p_i$. Then, given $\rho(p_i|x_i) \sim D(p_i|\alpha_i)$, we have $\rho(p_{ic}|x_i) \sim Beta(p_{ic}|\alpha_{ic}, \alpha_{i0} - \alpha_{ic})$, where $\alpha_{i0} = \sum_{k=1}^{K} \alpha_{ik}$. Thus, we further have:

$$
\rho(p_{ic}|x_i) = \frac{1}{B(\alpha_{ic}, \alpha_{i0} - \alpha_{ic})} p_{ic}^{\alpha_{ic}-1}(1 - p_{ic})^{\alpha_{i0}-\alpha_{ic}-1},
\tag{2}
$$

where $B(\cdot, \cdot)$ is the $K$-dimensional multinomial beta function, and $B(\alpha_{ic}, \alpha_{i0} - \alpha_{ic}) = \frac{\Gamma(\alpha_{ic})\Gamma(\alpha_{i0}-\alpha_{ic})}{\Gamma(\alpha_{ic}+\alpha_{i0}-\alpha_{ic})}$, $\Gamma(\cdot)$ denotes the gamma function. Based on this, we can further obtain the following

derivation:

$$
\begin{aligned}
P(y = c|x_i) &= \int p_{ic} \cdot \rho(p_{ic}|x_i) dp_{ic} \\
&= \int p_{ic} \cdot \left[ \frac{1}{B(\alpha_{ic}, \alpha_{i0} - \alpha_{ic})} p_{ic}^{\alpha_{ic}-1} (1 - p_{ic})^{\alpha_{i0} - \alpha_{ic} - 1} \right] dp_{ic} \\
&= \frac{B(\alpha_{ic} + 1, \alpha_{i0} - \alpha_{ic})}{B(\alpha_{ic}, \alpha_{i0} - \alpha_{ic})} \cdot \\
&\quad \int \frac{1}{B(\alpha_{ic} + 1, \alpha_{i0} - \alpha_{ic})} p_{ic}^{\alpha_{ic}} (1 - p_{ic})^{\alpha_{i0} - \alpha_{ic} - 1} dp_{ic} \\
&= \frac{B(\alpha_{ic} + 1, \alpha_{i0} - \alpha_{ic})}{B(\alpha_{ic}, \alpha_{i0} - \alpha_{ic})} \cdot 1 \\
&= \frac{\Gamma(\alpha_{ic} + 1)\Gamma(\alpha_{i0})}{\Gamma(\alpha_{i0} + 1)\Gamma(\alpha_{ic})} \\
&= \frac{\alpha_{ic}\Gamma(\alpha_{ic})\Gamma(\alpha_{i0})}{\alpha_{i0}\Gamma(\alpha_{i0})\Gamma(\alpha_{ic})} = \frac{\alpha_{ic}}{\sum_{k=1}^{K} \alpha_{ik}} = \frac{C_c(G(x_i))}{\sum_{k=1}^{K} C_k(G(x_i))} \\
&= \mathbb{E}[D(p_{ic}|\alpha_i)].
\end{aligned}
\tag{3}
$$

In our work, the outputs of classifier are adopting the exponential function. Thus, following [53], the outputs of DNNs in SAUE can be viewed as the expectation of Dirichlet distribution.

## C  Proof of Theorem 1

**Theorem 1** [37]. *Suppose we have given the m-th source data distribution $P_{\mathcal{S}_m}$, a hypothesis set $\mathcal{H}$, and a prior distribution $\pi$ over the hypothesis space $\Theta$. For any $\tau \in (0, 1]$ and $\lambda > 0$, with a probability at least $1 - \tau$ over the source samples $\mathcal{S}_m \sim P_{\mathcal{S}_m}^n$, for all posteriors $\rho$, we have:*

$$
\mathbb{E}_{\rho(\mathcal{H})}[\mathcal{L}(\mathcal{H})] \leq \mathbb{E}_{\rho(\mathcal{H})}[\tilde{\mathcal{L}}_{\mathcal{S}_m}(\mathcal{H})] + \frac{1}{\lambda} \left[ KL(\rho\|\pi) + \log \frac{1}{\tau} + \Psi_{\mathcal{S}_m, \pi}(\lambda, n) \right],
\tag{4}
$$

*where $\Psi_{\mathcal{S}_m, \pi}(\lambda, n) = \log \mathbb{E}_{\pi(\mathcal{H})} \mathbb{E}_{\mathcal{S}_m \sim P_{\mathcal{S}_m}^n} \left[ e^{\lambda(\mathcal{L}(\mathcal{H}) - \mathcal{L}(\tilde{\mathcal{H}}))} \right]$.*

**Lemma 1** [38]. *The PAC-Bayes bound, involving constants $\tau$ and $n$, as introduced in Theorem 1, is minimized by the Bayesian posterior $p(\mathcal{H})$, which represents the distribution over $\Theta$.*

*Proof.* The Donsker-Varadhan's change of measure states that for any measurable function $\phi : \Theta \to \mathbb{R}$, we have:

$$
\mathbb{E}_{\rho}(\mathcal{H}) \leq KL(\rho\|\pi) + \log \mathbb{E}_{\pi(\mathcal{H})}[e^{\phi(\mathcal{H})}].
\tag{5}
$$

Thus, with $\phi(\mathcal{H}) := \lambda(L(\mathcal{H} - \hat{L}(\theta, \mathcal{S}_m)$ and $\forall \rho$ over hypothesis space $\Theta$, we have:

$$
\begin{aligned}
\mathbb{E}_{\rho}(\mathcal{H}) \left[ \lambda(L(\mathcal{H}) - \hat{L}(\mathcal{H}, \mathcal{S}_m)) \right] &= \lambda \left( \mathbb{E}_{\rho(\mathcal{H})}[L(\mathcal{H})] - \mathbb{E}_{\rho(\mathcal{H})}[\hat{L}(\mathcal{H}, \mathcal{S}_m)] \right) \\
&\leq KL(\rho\|\pi) + \log \mathbb{E}_{\pi(\mathcal{H})}[e^{\lambda(L(\mathcal{H}) - \hat{L}(\mathcal{H}, \mathcal{S}_m))}].
\end{aligned}
\tag{6}
$$

For the non-negative random variable $\zeta_{\pi}(\mathcal{S}_m) := \mathbb{E}_{\pi(\mathcal{H})}[e^{\lambda(L(\mathcal{H}) - \hat{L}(\mathcal{H}, \mathcal{S}_m))}]$, we apply Markov's inequality on it, and have:

$$
\mathbb{P} \left( \zeta \leq \frac{1}{\tau} \mathbb{E}_{\mathcal{S}_m \sim P_{\mathcal{S}_m}^n} [\zeta_{\pi}(\mathcal{S}_m)] \right) \geq 1 - \tau.
\tag{7}
$$

This impiles that with probability at least $1 - \tau$ over the choice of $\mathcal{S}_m \sim P_{\mathcal{S}_m}^n$, we have $\forall \rho$ over hypothesis space $\Theta$:

$$
\mathbb{P} \left( \mathbb{E}_{\rho(\mathcal{H})}[\mathcal{L}(\mathcal{H})] \leq \mathbb{E}_{\rho(\mathcal{H})}[\hat{\mathcal{L}}_{\mathcal{S}_m}(\mathcal{H})] + \frac{1}{\lambda \left[ KL(\rho\|\pi) + \log \frac{1}{\tau} + \Psi_{\mathcal{S}_m, \pi}(\lambda, n) \right]} \right) \geq 1 - \tau,
\tag{8}
$$

where $\Psi_{\mathcal{S}_m, \pi}(\lambda, n) = \log \mathbb{E}_{\pi(\mathcal{H})} \mathbb{E}_{\mathcal{S}_m \sim P_{\mathcal{S}_m}^n} \left[ e^{\lambda(\mathcal{L}(\mathcal{H}) - \mathcal{L}(\tilde{\mathcal{H}}))} \right]$, and we prove the statement of the Theorem 1.

## D  Generalization Bound

**Lemma 2** [39]. *Suppose we have given the probability measures $\nu_{\mathcal{S}_m}, \nu_{\mathcal{T}} \in \mathcal{P}(\mathcal{F})$ of the $m$-th source feature $f_{\mathcal{S}_m}$ and the blended-target domain feature $f_{\mathcal{T}}$, a hypothesis space $\Theta$, and a subspace $\tilde{\mathcal{H}} \in \Theta$. Let $\mathcal{F}$ denote a fixed representation space and $c(f_{\mathcal{S}_m}, f_{\mathcal{T}})$ denote the adaptation cost. For the ideal classifier $h' \in \tilde{\mathcal{H}}$ and any classifier $h \in \tilde{\mathcal{H}}$ with $f_{\mathcal{S}_m} \sim \nu_{\mathcal{S}_m}$ and $f_{\mathcal{T}} \sim \nu_{\mathcal{T}}$, we have:*

$$|\epsilon_{\mathcal{S}_m}(h, h') - \epsilon_{\mathcal{T}}(h, h')| \leq \frac{1}{2} d_{\mathcal{H}\Delta\mathcal{H}}(\nu_{\mathcal{S}_m}, \nu_{\mathcal{T}}), \tag{9}$$

*where $\epsilon_{\mathcal{S}_m}$ and $\epsilon_{\mathcal{T}}$ denote the error on the $m$-th source domain and the error on the blended-target domain respectively, and $\epsilon_{\mathcal{T}} = \frac{1}{N} \sum_{j=1}^{N} \epsilon_{\mathcal{T}_j}$. $d_{\mathcal{H}\Delta\mathcal{H}}$ denotes the $\mathcal{H}\Delta\mathcal{H}$-distance.*

*Proof.* By the definition of $\mathcal{H}\Delta\mathcal{H}$-distance, we have:

$$\begin{aligned} d_{\mathcal{H}\Delta\mathcal{H}}(\nu_{\mathcal{S}_m}, \nu_{\mathcal{T}}) &= 2 \sup_{h, h' \in \mathcal{H}} |\mathrm{Pr}_{x \sim \nu_{\mathcal{S}_m}}[h(x) \neq h'(x)] - \mathrm{Pr}_{x \sim \nu_{\mathcal{T}}}[h(x) \neq h'(x)]| \\ &= 2 \sup_{h, h' \in \mathcal{H}} |\epsilon_{\mathcal{S}_m}(h, h') - \epsilon_{\mathcal{T}}(h, h')| \geq 2|\epsilon_{\mathcal{S}_m}(h, h') - \epsilon_{\mathcal{T}}(h, h')|. \end{aligned} \tag{10}$$

**Theorem 2.** *Based on Lemma 2, with the error of the ideal joint hypothesis $\eta' = \epsilon_{\mathcal{S}_m}(h') + \epsilon_{\mathcal{T}}(h')$ which is a sufficiently small constant, for any $\delta \in (0, 1)$, with probility at least $1 - \delta$, for every $h \in \mathcal{H}$, $\epsilon_{\mathcal{T}}(h)$ is bounded by the following terms:*

$$\epsilon_{\mathcal{T}}(h) \leq \epsilon_{\mathcal{S}_m}(h) + \frac{1}{2} \hat{d}_{\mathcal{H}\Delta\mathcal{H}}(\nu_{\mathcal{S}_m}, \nu_{\mathcal{T}}) + 4\sqrt{\frac{2d \log(2b') + \log(\frac{2}{\delta})}{b'}} + \eta', \tag{11}$$

*where $\eta' = \epsilon_{\mathcal{S}_m}(h') + \epsilon_{\mathcal{T}}(h')$ is the ideal error for the classifier, which is a sufficiently small constant. $b'$ is the size of unlabeled samples.*

*Proof.* From Lemma 2, we can obtain the following terms:

$$\begin{aligned} \epsilon_{\mathcal{T}}(h) &\leq \epsilon_{\mathcal{T}}(h') + \epsilon_{\mathcal{T}}(h, h') \\ &\leq \epsilon_{\mathcal{T}}(h') + \epsilon_{\mathcal{S}_m}(h, h') + |\epsilon_{\mathcal{T}}(h, h') - \epsilon_{\mathcal{S}_m}(h, h')| \\ &\leq \epsilon_{\mathcal{T}}(h') + \epsilon_{\mathcal{S}_m}(h, h') + \frac{1}{2} d_{\mathcal{H}\Delta\mathcal{H}}(\nu_{\mathcal{S}_m}, \nu_{\mathcal{T}}) \\ &\leq \epsilon_{\mathcal{T}}(h') + \epsilon_{\mathcal{S}_m}(h) + \epsilon_{\mathcal{S}_m}(h') + \frac{1}{2} d_{\mathcal{H}\Delta\mathcal{H}}(\nu_{\mathcal{S}_m}, \nu_{\mathcal{T}}) \\ &= \epsilon_{\mathcal{S}_m}(h) + \frac{1}{2} d_{\mathcal{H}\Delta\mathcal{H}}(\nu_{\mathcal{S}_m}, \nu_{\mathcal{T}}) + \eta' \\ &\leq \epsilon_{\mathcal{S}_m}(h) + \frac{1}{2} d_{\mathcal{H}\Delta\mathcal{H}}(\nu_{\mathcal{S}_m}, \nu_{\mathcal{T}}) + 4\sqrt{\frac{2d \log(2b') + \log(\frac{2}{\delta})}{b'}} + \eta'. \end{aligned} \tag{12}$$

Finally, the expected error on the blended-target domain can be bounded by utilizing the expected measures of NWD on the joint distribution of multiple source and blended-target domains.

## E  Additional Experiment Analysis

In this part, we present more visualization results with extra comparison methods including DANN [20] and AMDA [29]. All experiments are performed on task b+c→i+p of the ImageCLEF-DA [40].

**Distribution Analysis.** The t-SNE [54] feature visualization results of extra comparison methods are illustrated in Figure 1. Note that different color dots denote different domains. Compared to ResNet-50, due to the domain adversarial learning, DANN can better align the source and target domains. Furthermore, AMDA achieves better performance through multi-source and multi-target domain features. Due to style adaptation and uncertainty estimation and elimination, SAUE achieves the best performance. The features from the same class generated by SAUE are better clustered while those belonging to different classes are better separated.

**Confusion Matrix.** The comparison of confusion matrices with extra methods are illustrated in Figure 2. Although DANN and AMDA achieve significantly progress compared to ResNet-50,

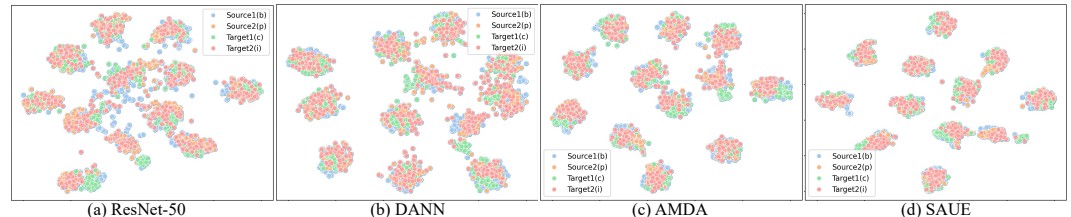

Figure 1: Visualization analysis of SAUE in task b+p→c+i. (Zoom in for clear visualization)

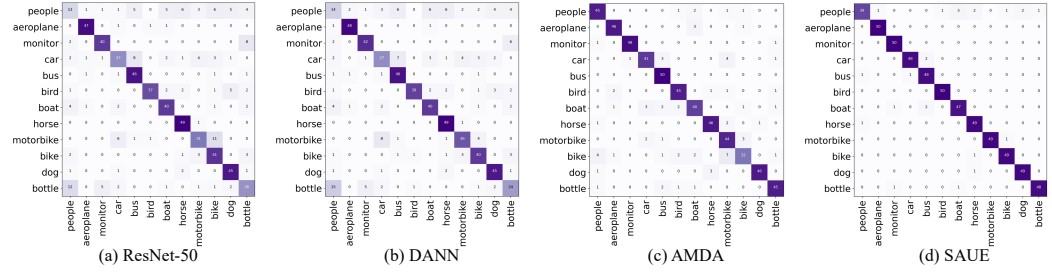

Figure 2: Confusion matrices of SAUE and comparison methods in task b+p→c+i. (Zoom in for clear visualization)

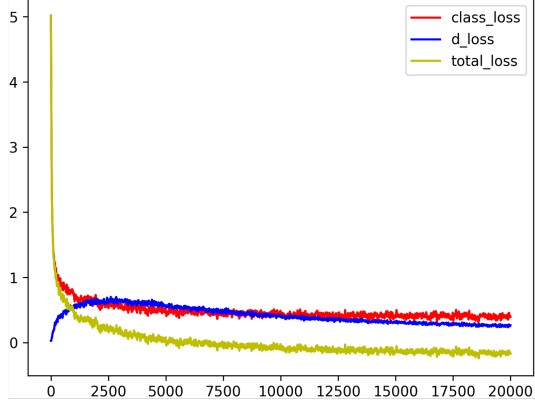

Figure 3: Loss functions with the increasing of iterations.

they still misclassified some classes, e.g., class "bike" is misclassified into class "motobike". In contrast, benefiting from the style adaptation and uncertainty estimation, SAUE generates more correct predictions which located on the main diagonal elements of confusion matrix.

**Convergence.** We further analyze the evolution of different loss functions with increasing iterations. The results are shown in Figure 3. The graphical representation illustrates that all the loss functions in our approach effectively converge as the training iterations increased. This convergence showcases the adaptability and reliability of our approach in MBDA tasks.

