# OpenReview forum: "Style Adaptation and Uncertainty Estimation for Multi-Source Blended-Target Domain Adaptation"
_NeurIPS.cc/2024/Conference — NeurIPS 2024 poster_

### Official Review · Reviewer_Sfz7 · 2024-06-22

**Soundness:** 3
**Presentation:** 4
**Contribution:** 3
**Rating:** 7
**Confidence:** 5

**Summary:**

This paper introduces a novel challenge in domain adaptation: the problem of Multi-Source Blended Target Domain Adaptation (MBDA). In MBDA, target domain distributions are blended, necessitating a model that can leverage features from multiple source domains to perform effectively on the blended-target domain. To address this problem, a Style Adaptation and Uncertainty Estimation (SAUE) approach is proposed. SAUE utilizes a similarity factor to select style information from the blended-target domain, creating a better representation space. Additionally, it enhances model robustness by employing a Dirichlet-based uncertainty estimation model to handle the diverse distributions introduced by multiple source domains. The effectiveness of the proposed method is validated through extensive image classification experiments. A theoretical analysis of SAUE underscores its potential in solving the MBDA problem.

**Strengths:**

1. The proposed MBDA problem is highly relevant for real-world applications. Unlike existing DA problems, MBDA incorporates confusing domain scenarios that mimic real-world situations where the source domain distributions are various, and the target domain distributions are blended. This assumption adds an additional challenge for DA, as the model needs to align domains without domain labels of the target domain.
2. The proposed method SAUE is innovative. The inclusion of style adaptation to enhance the source domain features is intriguing, where the similarity-based weighted matrix is cleverly used to select more useful target style information for constructing a better representation space.
3. The uncertainty estimation component effectively addresses the distribution discrepancy issue introduced by multi-source domains. In addition, the authors construct an adversarial learning strategy that aligns domains without the requirement of domain labels.
4. The paper is well-written with the theoretical analysis clearly articulated and thoroughly explained.

**Weaknesses:**

1. The paper mentions various terms related to domain adaptation in Sections 1 and 2, such as UDA, SSDA, MSDA, MMDA, MTDA, and BTDA. What are the main differences and relationships among these terms? The authors are encouraged to provide a table summarizing the main characteristics to distinguish these terms.
2. In the ablation study, the comparison primarily focuses on highlighting the improvements of each proposed module. It is suggested that the authors compare different feature augmentation methods to further emphasize the superiority of the proposed Style Adaptation module.
3. A minor point of interest is how the proposed method performs on other backbones, such as the Vision Transformer.

**Questions:**

Please refer to the Weaknesses part.

**Limitations:**

The authors have adequately claimed their limitations in Appendix B.

---

> ### Author Rebuttal · Authors · 2024-08-07
>
> Thanks for your positive affirmation and constructive comments. Below are our responses to the weaknesses and questions.
>
> >**W1:** The paper mentions various terms related to domain adaptation in Sections 1 and 2, such as UDA, SSDA, MSDA, MMDA, MTDA, and BTDA. What are the main differences and relationships among these terms? The authors are encouraged to provide a table summarizing the main characteristics to distinguish these terms.
>
> **A1:** We have summarized the differences and relationship among the mentioned domain adaptation problems in the PDF file (Table R5).
>
> >**W2:** In the ablation study, the comparison primarily focuses on highlighting the improvements of each proposed module. It is suggested that the authors compare different feature augmentation methods to further emphasize the superiority of the proposed Style Adaptation module.
>
> **A2:** We have introduced random augmentation (RA), MixStyle [a], and pAdaIN [b] techniques to further emphasize the superiority of the proposed SA module, please refer to the PDF file (Table R3 (a)).
>
> >**W3:** A minor point of interest is how the proposed method performs on other backbones, such as the Vision Transformer.
>
> **A3:** We have added experiments on the Office-Home dataset and utilized a new backbone, ViT. The results are illustrated on Tab. R6, which demonstrate that our method also effective when using the ViT backbone. We will add this comparison to the final version to further show the quality of our paper.
>
> Refs:
>
> [a] Domain generalization with mixstyle. ICLR 2021.
>
> [b] Permuted adain: Reducing the bias towards global statistics in image classification. CVPR 2021.

---

> ### Comment · Reviewer_Sfz7 · 2024-08-08
> **Thanks for the authors' responses**
>
> After thoroughly reviewing the questions raised by other reviewers and carefully considering the authors' responses, I am convinced that my concerns have been effectively addressed. I believe the proposed method is inspiring for real-world complex environments. As a result, I would like to raise my score.

---

> > ### Author Response · Authors · 2024-08-12
> > **Thanks for your support**
> >
> > Thank you for your exceptionally prompt feedback and unwavering support of our paper. We are glad to observe that our responses have effectively resolved your concerns. Your invaluable input has undeniably enhanced the quality of our paper, and we sincerely thank you for your dedication and time.

---

### Official Review · Reviewer_NQ8Y · 2024-07-09

**Soundness:** 3
**Presentation:** 3
**Contribution:** 2
**Rating:** 5
**Confidence:** 3

**Summary:**

This paper works on Multi-source Blended-target Domain Adaptation setting, which learns a model from multiple source domain and evaluates the model in a mixed multi-target domain without access to the domain labels of target data. This paper utilize the style information of the blended-target domain with weight factor to enhance source domain features for feature augmentation. To avoid the negative impact of domain-specific information in multi-source data, this paper also uses KL loss to reduce the impact of incorrectly classified source samples. This paper also proposes a domain adversarial loss with the help of nuclear-norm 1-Wasserstein discrepancy to reduce the domain gap between source domain and target domain by reusing the classifier of task network. Extensive experiments on various benchmark and comprehensive analysis prove the effectiveness of the proposed method.

**Strengths:**

1. This paper is well motivated.
2. The proposed method is evaluated with various benchmark and analyzed extensively.

**Weaknesses:**

1. The style adaptation method is not compared with prior works proposed for the same purpose like MixStyle.
2. Whether Domain Adversarial Alignment without Domain Labels outperform domain adversarial learning by treating multi-source data as one single source domain and multi-target data as one target domain is not discussed.

**Questions:**

Please see weakness part.

**Limitations:**

No limitation is discussed in the paper.

---

> ### Author Rebuttal · Authors · 2024-08-07
>
> Thank you very much for your feedback and questions. We provide our responses below
>
> >**W1:** The style adaptation method is not compared with prior works proposed for the same purpose like MixStyle.
>
> **A1:** We have compared our style adaptation (SA) with some prior works for the same purpose, such as random augmentation (RA), MixStyle [a], and pAdaIN [b]. The results are shown in Table R3 (a) on the PDF file. Compared to these prior works, our SA method effectively selects more suitable style information to augment the source features, achieving superior performance.
>
> >**W2:** Whether Domain Adversarial Alignment without Domain Labels outperform domain adversarial learning by treating multi-source data as one single source domain and multi-target data as one target domain is not discussed.
>
> **A2:** We have performed comparison experiments based on your advice (please refer to Table R4 in the PDF file). The experimental results show that the performance of domain adversarial alignment without domain labels outperforms the performance of domain adversarial learning by treating multi-source data as one single source domain and multi-target data as one target domain.
>
> Refs:
>
> [1] Domain generalization with mixstyle. ICLR 2021.
>
> [2] Permuted adain: Reducing the bias towards global statistics in image classification. CVPR 2021.

---

> > ### Author Response · Authors · 2024-08-13
> >
> > Dear Reviewer NQ8Y,
> >
> > We are truly thankful for your valuable time and constructive feedback!
> >
> > Since the discussion deadline is approaching (11:59pm AoE on August 13), we would like to kindly inquire if our rebuttal has addressed your concerns. We are willing to address any further issues you might have : )
> >
> > Sincerely,
> >
> > Authors

---

### Official Review · Reviewer_cWFB · 2024-07-13

**Soundness:** 3
**Presentation:** 3
**Contribution:** 2
**Rating:** 5
**Confidence:** 5

**Summary:**

The paper proposes a "Style Adaptation and Uncertainty Estimation (SAUE)" method for Multi-source Blended-Target Domain Adaptation (MBDA). The core objective of the SAUE method is to enhance the source domain features by leveraging style information from a blended-target domain, which is a mixture of multiple sub-target domains. The method employs a similarity factor to select beneficial target style information and integrates an uncertainty estimation technique to fortify the model's robustness. The proposed SAUE is evaluated on several domain adaptation benchmarks, including ImageCLEF-DA, Office-Home, and DomainNet, demonstrating superior performance over existing state-of-the-art techniques.

**Strengths:**

It is demonstrated an impressive results on several domain adaptation benchmarks with a significant improvement.
The paper is well-organized and easy to follow.

**Weaknesses:**

1. Although the blended-target domain adaptation has attracted much attention, it is similar to the open-compound domain adaptation that has no annotation of the target domain.  Authors should compare and discuss the difference between the open-compound domain adaptation works.

2. The proposed style adaptation and uncertainty estimation are motivated and effective ways to handle domain shift problems. However, the idea has been well studied in conventional domain adaptation [a], domain generation [b], and open-compound domain adaptation fields. Therefore, the proposed method seems to lack innovation.

    [a] Informative Data Mining for One-shot Cross-Domain Semantic Segmentation

    [b] Uncertainty modeling for out-of-distribution generalization

    [c] Open Compound Domain Adaptation with Object Style Compensation for Semantic Segmentation

3. In my opinion, the proposed style adaptation technique is not tailored for multi-source BTDA as it also works for single-source BTDA. Therefore, the statement "the first work for MBDA" seems to be weak.

**Questions:**

Detailed questions refer to the weaknesses.

**Limitations:**

The limitations of the work have been discussed.

---

> ### Author Rebuttal · Authors · 2024-08-07
>
> Thanks a lot for your efforts in reviewing the paper. Below, we respond to your questions in details.
>
> >**W1:** Comparison to OCDA.
>
> **A1:** The key characteristic of blended-target domain adaptation (BTDA) is that the target domain is a mixture of multiple sub-domains sharing the same category space. Compared to BTDA, open-compound domain adaptation (OCDA) handles a target domain consisting of multiple sub-domains, some known (open-set) and some unknown (compound-set). OCDA's challenge is adapting the model to perform well on both known and unknown sub-domains without annotations. In BTDA, the model learns domain-invariant representations generalizable across the blended-target domain. In OCDA, the model identifies and handles known sub-domains effectively while being robust to unknown ones. We will add this comparison and discussion in Section 2 of the final version.
>
> >**W2:** The innovation of SAUE.
>
> **A2:** IDM [a] introduced a style transfer technique to alleviate the confusion between some high-entropy prediction images and target images, they also utilized mean entropy and cosine similarity to estimate the uncertainty. However, IDM only considered the single source and single target domain scenario for semantic segmentation. DSU [b] inserted the uncertainty estimation of the feature means and feature deviation into an AdaIN module. However, DSU do not consider the more realistic scenario where the target domains are multiple and blended. OSC [c] constructed style compensation (SC) strategy mainly to augment the object feature for a semantic segmentation image by constructing the weight factor and discrepancy features. The weight factor of SC is calculated by representative-key and object features. However, OSC do not take model uncertainty and multiple source setting into account.
>
> Different from the previous methods [a], [b], and [c], in multi-source blended-target domain adaptation (MBDA), the distributions of source domains are diverse, while the target distributions are blended. Directly using style transfer techniques in the current BTDA scenario fails to capture domain-invariant representations among different source domains in MBDA. Our efficient and novel solution utilizes the Wasserstein distance to explicitly measure the similarity of low-level features between the source and target domains, thereby constructing a weight factor. We then use the weight factor to select target domain features that are more suitable for the source domains and use them for feature augmentation. In this way, our method can mitigate the impact of the domain-specific attributes. In addition, multiple source domains may generate multiple opinions in evidential deep learning (EDL) [d]. Thus, our method constructs an uncertainty estimation strategy that introduced a Dirichlet-based evidential model to fuse multiple opinions by using the Dempster-Shafer Rule, which is more beneficial to exploit valuable knowledge from multiple source domains.
>
> Although some methods (including the aforementioned three methods) introduced such techniques into semantic segmentation, there is no effective strategy to address the knowledge transfer between multiple source domains and a blended-target domain in a more complex and realistic MBDA scenario. To fully consider the above-mentioned issues, we proposed our style adaptation and uncertainty estimation method. Thus, compared most of the existed methods, the main innovations of our method can be summarized as: 1) the novel style adaptation strategy is designed to select suitable blended-target style information for feature augmentation of multiple source domains. 2) The uncertainty estimation strategy that utilized Dirichlet-based evidential model to fuse multiple opinions from multiple source domains, which more effectively exploit domain-invariant knowledge from multiple source domains. 3) As far as we know, our propose SAUE method is the first work that proposed for providing a novel solution for the challenging MBDA setting, which is a real and complex scenario. Meanwhile, we have provided a theoretical analysis and sufficient experimental validation of the proposed method.
>
> [a] Informative Data Mining for One-shot Cross-Domain Semantic Segmentation
>
> [b] Uncertainty modeling for out-of-distribution generalization
>
> [c] Open Compound Domain Adaptation with Object Style Compensation for Semantic Segmentation
>
> [d] Murat Sensoy, Lance Kaplan, and Melih Kandemir. Evidential deep learning to quantify classification uncertainty.
>
> >**W3:**  In my opinion, the proposed style adaptation technique is not tailored for multi-source BTDA as it also works for single-source BTDA. Therefore, the statement "the first work for MBDA" seems to be weak.
>
> **A3:** Although style adaptation works for BTDA, its integration within MBDA is designed to handle complexities unique existed in multi-source scenarios. Directly using target style information to augment source features can aggravate domain shift due to conflicts from diverse source domain distributions. Unlike previous style transfer modules, our style adaptation module selects suitable target style information for different source domains using the Wasserstein distance to construct a weight factor, creating a cohesive representation space and improving the adaptation process in a multi-source setting
>
> Moreover, there maybe some misunderstandings regarding our claim for “the first work for MBDA”. In the first point of our contributions, we state: “An approach SAUE is proposed to explore information from multiple source domains for BTDA. As far as we known, SAUE is the first work that proposed for MBDA, which can utilize more feature information from extra source domains to learn domain-invariant representations.” Therefore, we would like to clarify that the SAUE method is the first work specifically addressing MBDA not merely the utilization of style adaptation technique. We still appreciate the reviewer's detailed insights.

---

> > ### Author Response · Authors · 2024-08-13
> >
> > Dear Reviewer cWFB,
> >
> > We are truly thankful for your valuable time and constructive feedback!
> >
> > Since the discussion deadline is approaching (11:59pm AoE on August 13), we would like to kindly inquire if our rebuttal has addressed your concerns. We are willing to address any further issues you might have.
> >
> > Sincerely,
> >
> > Authors

---

### Official Review · Reviewer_LwRV · 2024-07-14

**Soundness:** 2
**Presentation:** 2
**Contribution:** 2
**Rating:** 6
**Confidence:** 4

**Summary:**

The paper addresses the setting of Blended target domain adaptation. The paper proposes style adaptation and uncertainty estimation for multi-source blended target domain adaptation. They propose to utilize the extra knowledge acquired from the blended target domain. Where a similarity factor is utilized to obtain target style information to augment the source features. During training, they utilize the information from the target domain with a scaling factor and Wasserstein distance between the source and target features. Moreover, they do not utilize the domain labels without which they claim to align the multi-source and blended target domains. They utilize common domain adaptation datasets to demonstrate the effectiveness of their method.

**Strengths:**

– The problem setting of addressing multi-target distribution shifts with a source trained model is useful

– Performance improvement is significant compared to the baseline methods.

**Weaknesses:**

The paper lacks extensive validation, clarity, and novelty. Elaborated below.

**Questions:**

– Can the authors demonstrate with the default version of DomainNet instead of utilizing a subset of it? Moreover, the inclusion of the VisDA dataset would also be useful.

– Could the authors also provide detailed numerical results of Figures C and D for better clarity?

– The idea of using the target and source features during the training process with a scaling factor and Wasserstein distance doesn’t seem novel. Moreover the paper doesn't cite [1][2] which makes the related work quite limiting.

– The idea of selecting style information is not new. Moreover, it is not clear which layer has been utilized for the style information and what the motivation is to utilize the style content at a particular layer.

– The distribution analysis can be moved to the appendix, which doesn’t offer significant insights of the method.

– Akin to MCDA [12], could the authors utilize ResNet 101 instead of ResNet50 and compare the results for the default version of the DomainNet dataset? With powerful backbones, it would be interesting how the style features are still preserved.

– Extensive implementation details have not been provided; the batch size of the blended target domain, learning rates, and which layers from the ResNet model have been utilized for this if their code will be released. Does the method necessitate a high batch size for the target domain? In this case, it might be a limitation.

References:
[1] Shen, Jian, et al. "Wasserstein distance guided representation learning for domain adaptation." Proceedings of the AAAI conference on artificial intelligence. Vol. 32. No. 1. 2018.
[2] She, Qingshan, et al. "Improved domain adaptation network based on Wasserstein distance for motor imagery EEG classification." IEEE Transactions on Neural Systems and Rehabilitation Engineering 31 (2023): 1137-1148.

**Limitations:**

Yes.

---

> ### Author Rebuttal · Authors · 2024-08-07
>
> We appreciate the comments and questions you post. Here are our point-to-point responses.
>
> >**Q1:** Experiments on the DomainNet and VisDA datasets.
>
> **A1:** Due to time limitations, we provide comparisons with important and recent multi-source blended-target domain adaptation (MBDA) methods MCDA and DGWA on the DomainNet and VisDA datasets. Initial experiments are shown in Tables R1 and R2 (see PDF), which demonstrate the superiority of our SAUE method on these challenging datasets. We will give more comparisons in the revised version.
>
> >**Q2:** Figs. 2C and 2D.
>
> **A2:** Detailed numerical results for Figures 2C and 2D are provided for clarity (see Table R7 in the PDF).
>
> >**Q3:** Idea about feature learning and related works.
>
> **A3:** Most of the existing methods use target and source features based on the single source and single target domain adaptation (SSDA) or single source and blended target domain adaptation (BTDA), which do not consider MBDA setting. In MBDA, challenges include: 1) various distributions from multiple source domains may introduce potential conflicts. 2) The target distributions are mixed. 3) Due to different discrepancies between different source and target domains, it is challenging to align multiple source and blended-target domains simultaneously. Our efficient and novel solution is to utilize the Wasserstein distance to explicitly measure the similarity of low-level features between the source and target domains to construct a weight factor.
>
> The innovations of our proposed method are: 1) a novel style adaptation strategy that selects suitable blended-target style information for feature augmentation of multiple source domains. 2) An uncertainty estimation strategy that utilized Dirichlet-based evidential model to fuse multiple opinions from multiple source domains, which more effectively exploit domain-invariant knowledge from multiple source domains. 3) The proposed SAUE method provides a novel solution for the challenging MBDA setting.
>
> We have cited and discussed these two good methods [1], [2]. The revised parts, which will be added to the final version, are as follows: WDGRL [1] employs a neural network to estimate the Wasserstein distance between the source and target domains, optimizing feature representations to minimize this distance. Shen et al. [2] introduced an improved domain adaptation network for motor imagery (MI) classification and utilized Wasserstein distance to construct a domain adversarial learning strategy to handle EEG-based MI detection tasks.
>
> >**Q4:** Style information selection and motivation.
>
> **A4:** While the technique of selecting style information has been explored in other contexts, our contribution lies in the unified adaptation framework designed for MBDA. Given the challenge that different discrepancies exist between different source and target domains, directly utilizing feature augmentation techniques like random augmentation may aggravate the domain shift problem. Therefore, we designed a strategy that selects style information to augment the source domains’ features. Our approach constructs tailored style information selection with a novel similarity-based weighted matrix strategy for MBDA, effectively selecting suitable blended-target domain style information for specific source domain, showing empirical gains and component synergy.
>
> The style information is extracted from the second bottleneck layer of ResNet and utilized in the third bottleneck layer. The motivation for utilizing style content at a particular layer is to leverage the style information of the blended-target domain to augment the features of the source domains. Previous work [3] has demonstrated that low-level features (e.g., intermediate features from the second bottleneck of ResNet) of deep neural networks (DNNs) primarily represent style information. Building on this insight, we extract the style information of the blended-target domain from the low-level features of ResNet and use it to augment the features of the source domains.
>
> >**Q5:** Distribution analysis.
>
> **A5:** We will move the distribution analysis part to the appendix in the revised paper.
>
> >**Q6:** Evaluation of different backbones.
>
> **A6:** We have analyzed the performance of our method using the ResNet-50 and ResNet-101 backbones in the default version of the DomainNet dataset (please see Table R2 in the PDF file). The experimental results show that our method achieves significant performance gains when utilizing powerful backbones (ResNet-101). We compared the performance gains of the style adaptation module with standard backbone (ResNet-50) and powerful backbone (ResNet-101). The results indicate that the performance gains (+2.9% and +2.3%) of the style adaptation module when integrated into the ResNet-50 backbone surpass those (+2.3% and +1.3%) when integrated into the ResNet-101 backbone.
>
> >**Q7:** Implementation details and batch size.
>
> **A7:** For better understanding, we provide more implementation details as follows. The batch size for the blended-target domain is 32, same as the source domains. The learning rate starts at 1e-3, updated by the LambdaLR strategy. We use an ImageNet pre-trained ResNet, replacing the last FC layer with task-specific FC layers. Our method does not require a high target domain batch size, only equal to that of the source domains. Extra validation experiments (see Table R3(b) in the PDF) show different target domain batch sizes do not significantly impact performance. Thus, the batch size for blended-target domain is not a limitation.
>
> Thanks again for your meaningful questions.
>
> Refs:
>
> [1] Wasserstein distance guided representation learning for domain adaptation. AAAI 2018.
>
> [2] Improved domain adaptation network based on Wasserstein distance for motor imagery EEG classification. IEEE TNSRE 2023.
>
> [3] Mixstyle neural networks for domain generalization and adaptation. IJCV 2024.

---

> > ### Comment · Reviewer_LwRV · 2024-08-12
> >
> > Thank you for your efforts. My comments have been addressed. I would encourage the authors to include DomainNet (345 classes) and additional comparisons to it in the main paper, as well as discussions about feature learning and style information and additional implementation details in the manuscript. In response, I have increased my score.

---

> > > ### Author Response · Authors · 2024-08-12
> > > **Thanks for your support**
> > >
> > > Thank you for your active involvement and prompt feedback during the discussion phase. We are pleased to know that our rebuttal has effectively addressed your questions. We will carefully incorporate all of your valuable suggestions into the final version of our paper, including results on DomainNet (345 classes), discussions on feature learning and style information, as well as additional implementation details. Your constructive review has undoubtedly enhanced the quality of our paper, and we sincerely appreciate your dedication and time.

---

### Author Rebuttal · Authors · 2024-08-07

Dear Reviewers and Area Chair,

We sincerely thank all the reviewers for their positive comments and helpful feedback, which have significantly improved the quality of this paper. We have uploaded the responses to each reviewer along with the one-page PDF.

In response to the comments, we have carefully revised and enhanced the manuscript with the following additional discussions and experiments:

1.	Additional experiments on the default version of the DomainNet dataset and the VisDA-2017 dataset.
2.	Comparison of different backbones (ResNet-50, ResNet-101, and ViT-B/16).
3.	Detailed numerical results for Figures 2C and 2D.
4.	More implementation details about the proposed method.
5.	Discussion about the related work.
6.	Comparison of different domain adaptation settings.
7.	Comparison of different feature augmentation methods.

We hope our response sincerely addresses all the reviewers’ concerns.

Thank you very much for your time and consideration.

Best regards,

Submission3121 Authors

---

### Decision · Program_Chairs · 2024-09-25

**Decision:**

Accept (poster)

**Comment:**

This paper presents a Style Adaptation and Uncertainty Estimation approach to tackle the Multi-Source Blended Target Domain Adaptation problem, where target domain distributions are blended. During the rebuttal phase, the authors provided additional experiments and discussions on challenging backbones, datasets, and difficult UDA settings. Following the response and multiple rounds of discussion, the paper received four positive scores, including one accept, one weak accept, and two borderline accept. Considering the reviewers'  positive comments and the paper's contributions, the meta-reviewer gives the final recommendation of acceptance.